# A History and Atlas of the Human CD4^+^ T Helper Cell

**DOI:** 10.3390/biomedicines11102608

**Published:** 2023-09-23

**Authors:** Jacqueline M. Crater, Daniel C. Dunn, Douglas F. Nixon, Robert L. Furler O’Brien

**Affiliations:** Division of Infectious Diseases, Department of Medicine, Weill Cornell Medicine, 413 E 69th St., Belfer Research Building, New York, NY 10021, USA

**Keywords:** CD4^+^ T cell, primary immunodeficiencies, cell migration, adaptive immunity

## Abstract

CD4^+^ T cells have orchestrated and regulated immunity since the introduction of jawed vertebrates, yet our understanding of CD4^+^ T cell evolution, development, and cellular physiology has only begun to be unearthed in the past few decades. Discoveries of genetic diseases that ablate this cellular population have provided insight into their critical functions while transcriptomics, proteomics, and high-resolution microscopy have recently revealed new insights into CD4^+^ T cell anatomy and physiology. This article compiles historical, microscopic, and multi-omics data that can be used as a reference atlas and index to dissect cellular physiology within these influential cells and further understand pathologies like HIV infection that inflict human CD4^+^ T cells.

## 1. Evolution of the CD4^+^ Helper T Cell

Prior to the advent of the CD4^+^ T cell and other components of the adaptive immune system, innate immunity provided life with powerful means to protect themselves. Early animals evolved specialized innate cells that can phagocytize foreign material, recognize common microbial pathogen-associated molecular patterns, and induce cytolysis to protect the host [1,2,3]. Antibiotics, complement, exotoxins, RNAi, and CRISPR-Cas9 are among additional countless mechanisms of innate protective chemicals that are synthesized to repel competitive lifeforms and viruses. Interleukins and their receptors far predate the emergence of lymphocytes, some of which can be identified as early as *Trichoplax* [4]; however, invertebrates lack critical adaptive immune lymphoid organs, such as lymph nodes, thymus, and spleen [1,5]. 

Although these innate immune protections are quite effective, viruses and bacteria replicate and mutate at an exponential rate compared to higher multicellular animals, providing an evolutionary advantage to highly replicative microbes that can mutate their genetic code. Although invertebrates lack adaptive immunity, jawless vertebrates such as hagfish and lampreys possess variable lymphocyte receptors (VLRs) with limited mutation capabilities that are precursors to antigen receptors found in jawed vertebrates [6,7]. An evolved system of adaptive immunity, resulting in specific and diverse antigen receptors, evolved over 400 million years ago in jawed vertebrates or *Gnathostomes* [8,9]. All four TCR genes (α,β,γ,δ) and the RAG1 and RAG2 recombinases required for somatic hypermutation are only found in jawed vertebrates [10]. T and B lymphocytes within these animals gained the ability to hypermutate antigenic B cell and T cell receptors (BCRs and TCRs) at specific genomic loci, providing these animal lineages a key protective advantage in the war against pathogenic microbes. 

The protective capabilities of newly evolved adaptive immunity are clear; however, the root cause of its introduction to multicellular life is still undefined. One theory postulates that animals with jaws may have ingested food that damages the gut and makes the animal susceptible to infections, and the emergence of adaptive immunity combats this risk [1]. Another hypothesis proposes that invertebrates simply lack the vasculature and lymphoid organs, such as lymph nodes, thymus, and spleen required to support adaptive immunity [1,5,11]. The RAG1 and RAG2 recombinases required for somatic hypermutation are thought to be derived from retroviral transposases [12], indicating a viral infection may have played a role in the evolution of adaptive immunity. Although the answer to how adaptive immunity arose is still under debate, the simultaneous presence of B cells, CD8^+^ T cells, and CD4^+^ T cells is highly evolutionarily conserved from early jawed vertebrates to humans [6]. The simultaneous evolutionary presence of the CD4^+^ helper T cell alongside effector B and CD8^+^ T cells emphasizes their importance in adaptive immunity; however, their discovery remained elusive until the 19th century.

## 2. Discovery of the CD4^+^ Helper T Cell

CD4^+^ T cells, or “helper” T cells, are now regarded as an integral part of the adaptive immune response. However, only after multiple human clinical reports and murine studies did the field begin to appreciate how T cells develop and influence the immune response. 

The thymus is a primary lymphoid organ where T cells differentiate from common lymphoid progenitors. Before the 1960s, many physicians and investigators regarded the thymus as an insignificant organ. However, in 1961, Jacques Miller observed that mice thymectomized directly after birth were more susceptible to infection but did not reject foreign tissue grafts [13]. Miller proposed that the thymus must be implicated in lymphocyte development in neonates. Miller also noted that his thymectomized mice produced minimal germinal centers within lymph nodes [13]. Thymus-derived cells were now linked to protection against infection, rejection of transplants, and stimulating humoral immunity. These experiments provided evidence that the thymus was significant and led to the labeling of thymus-derived lymphocytes as “T” cells (Figure 1).

Research into the initiation of humoral immunity, such as the function of B plasma cells to produce antibodies [14], paved the way to elucidate the roles of thymus-derived lymphocytes. Irradiated mice treated with rat erythrocytes were injected with both bone marrow-derived and thymus-derived cells. Following these injections, subject antibody responses were larger in the combined condition than in either condition alone [15]. This synergistic effect indicated an interaction between the bone marrow-derived and thymus-derived lymphocytes [15]. The bone marrow-derived cells, now referred to as B cells (B originated from *Bursa of Fabricius* in birds), produced antibodies, while the thymus-derived cells were necessary to initiate and “help” antibody production [15]. While T cell help was their first established function, another function of T cells was soon uncovered—cytotoxicity [16]. Like B cells, cytotoxic T cells were found to be dependent on interactions with helper T cells [17].

The function of the T cells as “helpers” or cytotoxic “killers” was then tied to cell surface antigen expression using monoclonal antibodies. OKT4 was found to be reactive with helper T cells [18], while the OKT8 antibody was found to label cytotoxic T cells [19]. A third monoclonal antibody OKT3 was found to be reactive with both T cell subsets and is a useful marker for distinguishing T cells from other cells within the body [20]. These cell surface antigens were denoted as T4, T8, and T3, respectively, which were renamed to clusters of differentiation (CD) CD4, CD8, and CD3. In 1981, the first reported cases of Acquired Immunodeficiency Syndrome (AIDS) propelled intense further investigation into the function and characterization of the CD4^+^ T cell subset.

In 1991, CD4^+^ T cells were divided into two subsets based on their helper responses. The helper T cells that initiate cytotoxic T cell responses and produce IL-2 and IFN-γ were termed Th_1_ [21]. The helper T cells responsible for stimulating antibody responses and producing IL-4 and IL-5 were termed Th_2_ cells [21]. This has been followed by the identification of additional CD4^+^ T cell subsets: T_reg_ [22], Th_17_ [23], Th_9_ [24], T_fh_ [25], and Th_22_ [26] in addition to multiple temporal states of differentiation including naïve, effector, and memory (central, effector, and resident). Others have reported a cytotoxic role for a subset of CD4^+^ T cells that may express CRTAM, CD38, NKG2D, SLAMF7, and CX3CR1 [27]. Although a clearly defined set of surface markers remains to delineate cytotoxic CD4^+^ T cells from other CD4^+^ T cell subsets, there is a growing interest in the role of this cytotoxic subset in fighting infectious diseases, promoting autoimmunity, and eliminating cancers. More in-depth analysis of the functional characteristics of each helper T cell subset and the history of their discoveries can be found in previous reviews [27,28,29]. This increasing spectrum of T cells helps emphasize the broad role of CD4^+^ T cells in orchestrating and balancing the powerful immune response.

## 3. The Importance of CD4^+^ T Cell Help in Orchestrating and Balancing the Power of Immunity

CD4^+^ T cells orchestrate the effector responses of B cells and CD8^+^ T cells, which thwart pathogens daily. However, these destructive capabilities must be kept in check to prevent allergies, autoimmunity, and transplant rejections. During development, autoreactive T and B cells are typically eliminated. Additionally, a two-step antigen verification system has evolved to fine-tune immune responses. In the first step, B and CD8^+^ T cells recognize foreign antigens presented by myeloid antigen-presenting cells. In the second step, a signal is provided by a CD4^+^ helper T cell that also recognizes the foreign antigen. This second signal dictates whether the effector cell will proliferate, differentiate, or undergo apoptosis. CD4^+^ T cells impact dendritic cell-mediated expansion of CD8^+^ T cells (CTL) in vitro. Full antigen-specific CD8^+^ T cell expansion occurs when both CD4^+^ and CD8^+^ T cells are present on a given dendritic cell, suggesting the 3-cell interaction theory has significant merit [30]. Oftentimes, this secondary signal includes TNF receptor stimulation and secreted cytokines provided by the CD4^+^ T cell. If the B or CD8^+^ T cell inappropriately recognizes host antigens, a second signal will not be given by CD4^+^ T cells. If the immune system is educated appropriately, a regulatory CD4^+^ T cell (T_reg_) may provide anti-proliferative or pro-apoptotic signals to the potentially damaging effector cell. When CD4^+^ T cell help is absent, dysfunctional, or hyperactive, this delicate balance is thrown off, leading to increased infections, autoimmunity, or allergies (Figure 2).

Autoimmunity arises when self-antigens are inappropriately targeted due to dysfunctional T cell help. In patients with rheumatoid arthritis, CD4^+^ T cells activate B cells release of cytokines that recruit innate immune cells to joints [31]. Patients with systematic lupus erythematosus (SLE) have increased circulating T_fh_ cells, which support auto-antibody-producing B cells [32]. Th_17_ cells play a large role in autoimmune diseases like rheumatoid arthritis, SLE, and multiple sclerosis [33]. The increased production of IL-17 by these CD4^+^ T cells helps activate auto-antibody-producing B cells and recruits neutrophils to the site of inflammation [33]. This inappropriate pro-inflammatory T cell help is typically hindered by regulatory T cells (T_regs_).

While many CD4^+^ T cell subsets promote inflammation, T_regs_ have immune-dampening capabilities. Too few or too many T_regs_ can shift immune responses towards autoimmunity, allergy, infections, transplant rejections, and cancer. In normal immune homeostasis, CD4^+^ T_regs_ prevent chronic immune activation and maintain immune tolerance of healthy tissues. Downregulation of T_regs_ in autoimmunity leads to autoreactive effector cells to go unchecked [34,35]. Clinical trials testing therapeutic T_reg_ infusions for autoimmune diseases including type 1 diabetes and Crohn’s disease are currently being explored [34]. For a more comprehensive review of CD4^+^ T cell subsets in autoimmunity, please see Rafael et al. [33]. Although too little regulatory help leads to allergies and autoimmunity, increased activity of CD4^+^ T_regs_ can prevent protective anti-cancer or anti-infection immune responses that can worsen patient survival [36,37]. This Goldilocks nature of CD4^+^ T cell help is highlighted by the multiple opportunistic infections, allergies, and autoimmunity that arise in patients following ablation or dysfunction of the CD4^+^ T cell compartment.

## 4. The Loss of Efficient CD4^+^ T Cell Help through Pathophysiological Migration and Activation

When components of the immune response are hindered or missing, immunodeficiencies can occur, which can be divided into two broad categories. Primary immunodeficiencies (PIDs) are typically caused by genetic mutations that are present from birth. Secondary immunodeficiencies are acquired from pathogens, chemotherapy, radiation, or other environmental factors. Here, we briefly discuss primary and secondary immunodeficiencies that directly impact CD4^+^ T cells to highlight the importance of T cell help. 

Mutations affecting T cell development, activation receptors, or migration can all result in primary immunodeficiencies involving T cell help. Table 1 outlines primary T cell immunodeficiencies, but as new mutations are continually discovered, this table is not exhaustive. Although the table below lists several mutations that lead to CD4^+^ T cell dysfunction or depletion, there are several more PIDs reviewed elsewhere that also affect other components of the immune system [38,39].

Aside from genetic predispositions, CD4^+^ T cell number and function can be disrupted by environmental events, causing acquired immunodeficiencies. Perhaps the most well-known acquired immunodeficiency is AIDS (Acquired Immunodeficiency Syndrome). AIDS is caused by HIV-1 (Human Immunodeficiency Virus-1) infection, and is characterized by the progressive loss of CD4^+^ T cells. HIV-1 infects CD4^+^ T cells via the CD4 receptor and the coreceptors CCR5 or CXCR4. Over the course of untreated HIV-1 infection, CD4^+^ T cell levels become depleted [64]. Without T cell help, the body becomes susceptible to deadly opportunistic infections and increased allergies and autoimmunity. AIDS is defined as an HIV-1 induced depletion of peripheral blood CD4^+^ T cells below 200 cells/μL or due to the presence of an AIDS-defining illness (ADI) [65] including persistent pneumonia, tuberculosis, or Kaposi’s sarcoma [65,66,67].

The loss of CD4^+^ T cells following primary or secondary immunodeficiencies has highlighted the critical importance of T cell help in preventing allergies, autoimmunity, and opportunistic infections. The genetic determinants of several of the primary immunodeficiencies above relate to the actin cytoskeleton. To understand why these actin regulatory proteins play such a large role in CD4^+^ T cell function and viability, it is important to appreciate how actin dictates CD4^+^ T cell migration and immune synapse formation. 

## 5. The Nomadic Life of a CD4^+^ Helper T Cell

T cells go through cyclical rounds of migration, proliferation, and differentiation throughout their development. Prior to becoming T cells, hematopoietic stem cells (HSCs) localize with osteoblasts within the bone marrow [68,69]. As chemokines and cytokines signal for HSCs to differentiate into common lymphoid progenitors (CLPs), these multipotent cells vacate the bone marrow niche through the peripheral blood and move to the thymus [69,70]. The CLPs then traverse the cortico-medullary junction and migrate to the thymic cortex parenchyma [71]. In the thymus, CLPs receive further signaling from thymic epithelial cells to activate the genes to create RAG1 and RAG2 enzymes. These recombinases induce the creation of antigen-specific T cell receptors (TCRs) on the cell surface. Thymic signaling also induces CD4 and CD8 gene expression. Thymic cells select for double positive (DP) CD4^+^CD8^+^ T cells by screening for their ability to bind to MHCs, a process known as positive selection [72] which induces these DP thymocytes to survive and differentiate further. DP thymocytes that are autoreactive are removed by negative selection [73], resulting in a population of self-tolerant naïve T cells.

Mature naïve CD4^+^ T cells travel up an S1P gradient and exit the thymus at the corticomedullary junction to enter the peripheral blood [74]. These cells then circulate through the peripheral blood and secondary lymphoid organs where they encounter antigen-presenting cells (APCs), such as dendritic cells, that potentially present their cognate antigen [75,76]. 

The secondary lymphoid tissue microenvironment has many different cues and signals that affect T cell differentiation during their sequential interactions with antigen-presenting dendritic cells [75]. Distinct subsets of antigen-presenting dendritic cells have differential localization within secondary lymphoid tissues. Dendritic cell localization influences specific T cell responses and sequential interactions between dendritic cells and T cells may occur. In T cell zones, the chemokines CCL19 and CCL21 attract CCR7^+^ dendritic cells that may be involved in maintaining T cell homeostasis or the activation of Th_1_ effectors during infection. CXCR5^+^ dendritic cells, which are found in the subcapsular sinus and perifollicular areas of the lymph node, may be required for the activation and development of Th_2_ cells.

Naïve T cells once activated, begin to proliferate and differentiate into effector CD4^+^ T cells [77]. Although differentiated follicular helper T cells will only migrate a small distance to help with B cell antibody production within nearby lymph node follicles, other helper T cells will egress from the lymphatics, reach the circulation, and then enter inflamed peripheral tissues like the gut and lungs to aid macrophage or cytotoxic T cell activation. Whichever the downstream effector site is, the T cell will once again adhere to an appropriate target cell and carry out its differentiated function. Following the resolution of inflammation, many of the terminally differentiated effector cells will undergo apoptosis while others will be retained within the peripheral tissues as resident memory T cells (T_RM_).

Migration and immune synapse formation are critical determinants of CD4^+^ T cell function and are highly reliant on the dynamic actin cytoskeleton. In PIDs that affect the actin cytoskeleton (Table 1), these genetic mutations frequently affect both migratory and cell activation signaling cascades. Both linear and branched actin structures are required at the lamellipodia and in building a strong immune synapse. When formins, ARP2/3, or their regulators are dysfunctional or absent, pathophysiological activation and migration ensue. The underlying physiology that allows for T cell migration, activation, and differentiation can be investigated with a more thorough understanding of T cell anatomy and protein expression during different stages of development.

## 6. Anatomy of a CD4^+^ Helper T Cell

While many images of T cells within peripheral blood smears depict spherical cells with large nuclei and scanty cytoplasm, this is not an accurate representation of T cells migrating or carrying out effector functions within tissues. Migrating T cells exhibit a polarized morphology that is controlled by organelle and biochemical pathway localization at distinct subcellular compartments. An extensive review by Niggli details several localized signaling and structural details of T cells [53]. 

Throughout this paper, we show scanning electron microscopy (SEM) and transmission electron microscopy (TEM) images of human CD4^+^ T cells that display the polarized structure of migrating cells. The polarized cytoskeleton of a migrating CD4^+^ T cell allows for the compartmentalization of organelles and biochemical reactions (Figure 3 and Appendix A). At the leading edge, actin-rich protrusions termed lamellipodia and filopodia extend and retract to guide cell movement following signaling through chemokine receptors and integrins along with other adhesion molecules [78]. The cell shown in Figure 4 turns to the left, as seen by the extension of the lamellipodia on the left side and the retraction of protrusions on the right side. This dynamic actin leading edge is directly connected to the nucleus, which takes up a large volume within the anterior of the cell. Figure 5 illustrates the different segments of a migrating CD4^+^ T cell. There is a general absence of organelles in the leading edge with organelles concentrated in the rear of the cell. The actin at the leading edge is tethered to the nucleus cell by LINC (linker of nucleoskeleton and cytoskeleton) complexes, which cross the nuclear envelope and connect chromatin to actin, microtubules, and intermediate filaments [79]. Posterior to the nucleus, the microtubules and vimentin intermediate filaments provide the cytoskeletal scaffold of the trailing tail of the cell, which has been coined the uropod. Electron microscopic examination of the uropod indicates there are at least two distinct portions of the uropod delineated by the centrosome, which we refer to as the proximal and distal uropod (Figure 4 and Figure 5). Although often overlooked in the literature as a separate region, the distal uropod has been referred to as the uropod “knob” in a few reports [53,80]. In SEM micrographs, the plasma membrane of the proximal uropod is smooth as opposed to the ruffled and filopodia-rich regions of the lamellipodium and distal uropod, respectively. TEM micrographs indicate that the centrosome organizes microtubules which anchor organelles in both the proximal and distal uropod (Appendix A). 

The location of organelles within a migrating T cell is not random and is informed by the organelle’s function. Vesicles may be released at the leading edge when recycling chemokine receptors or adhesion receptors, but multivesicular bodies and lysosomes are often located within the distal uropod. Although mitochondria are often located closer to the nucleus, these can extend to distal uropod. The three-dimensional localization of mitochondria may allow for different biochemical reactions to occur at the front and rear of the cell; however, this hypothesis needs to be tested. While CD4^+^ T cells have slightly different organellar organization and numbers depending on differentiation and activation state, this general anatomical organization underlies all migrating CD4^+^ T cells. 

Comparing healthy and disrupted CD4^+^ T cell morphology helps expand our understanding of disease pathology and progression. Several primary T cell immunodeficiencies are caused by mutations in proteins that control cytoskeletal rearrangements, preventing T cells from migrating towards sites of inflammation and infection or initiating an effective immune synapse (see Table 1). CD4^+^ T cell morphology is also disrupted during secondary immunodeficiencies like HIV-1 infection due to Nef-mediated cytoskeletal rearrangements [81,82]. HIV-1, like other enveloped viruses, also takes advantage of cortical cytoskeletons to optimally infect cells and produce virions [83]. As a CD4^+^ T cell migrates, HIV-1 enters the cell using CD4 and chemokine receptors at the leading edge. HIV-1 virions form at the distal uropod of migrating T cells, taking advantage of localized cholesterol-enriched lipid rafts and budding machinery.

With an understanding of the gross anatomy of the migrating CD4^+^ T cell, one can begin to dissect normal cellular physiology. But to delve deeper into discerning cellular pathologies following primary and secondary immunodeficiencies or T cell lymphomas, it is important to map out which of the twenty plus thousand genes and proteins are temporally expressed during each stage of T cell maturation. This gene and protein atlas, combined with high-resolution and time-lapse imaging, can provide additional insight into how these cells normally function and how they are affected during disease states.

## 7. Transcriptomic and Proteomic Atlas of the Human CD4^+^ Helper T Cell

Helper CD4^+^ T cell function is determined by distinct intracellular localization of subset-specific proteins. Advancing Omics technologies have identified these subset-specific proteins while high resolution imaging allows us to map spatial and temporal protein expression within human CD4^+^ T cells. Analysis of publicly-available transcriptomics datasets and our current proteomics dataset enables us to identify which of the 20,000+ genes are transcribed and translated in human CD4^+^ T cells at different stages of activation. We have annotated one public transcriptomic dataset (Appendix A) that identifies differential gene expression before and at three time points after activation. This temporal dataset delineates genes that are expressed during the transition of quiescent migratory to proliferative phases of the CD4^+^ T cell program.

This key dataset measures temporal differentially expressed genes originating from three different populations of quiescent human CD4^+^ T cells: Naïve (T_n_ = CD45RA^+^CD27^+^), Central Memory (T_cm_ = CD45RO^+^CD27^+^), and Transitional Memory (T_tm_ = CD45RO^+^CD27^−^) sorted from peripheral blood [84]. Bulk RNA-seq was performed before activation and after 40, 150 min, and 15 h of stimulation with anti-CD3 and anti-CD28 antibodies. Patterns of gene upregulation or downregulation during the transition from quiescent migratory to proliferative phases could be discerned for a large subset of the expressed genes. In Appendix A, expressed genes are arranged into cytoskeletal, metabolic, subcellular location, and physiological programs that change during the transition from a quiescent to an activated proliferative phase. This index of genes is useful in narrowing down which genes are likely needed during migratory or proliferative phases in the CD4^+^ T cell life cycle. 

To further characterize which proteins are translated in an expanded post-activation effector phase of CD4^+^ T helper cells, we performed mass spectrometry on cells from six separate healthy donors. Previously activated and expanded CD4^+^ T cells were permeabilized and proteins were precipitated using acetone before in solution trypsin digestion was performed, followed by stage tip desalting and LC MS/MS. Each sample was analyzed using a data independent acquisition (DIA) method. The data were searched against customized database containing Uniprot human protein sequences. Approximately five thousand human proteins were present in large enough quantities to be measured in our samples. These proteins are categorized within the table according to their cellular localization and protein function (Appendix A).

The use of these expression atlases can be used to narrow down potential gene/protein candidates of interest when studying diseases or functions of primary human CD4^+^ T cells. These datasets, in combination with other publicly available datasets, can be referred to when designing experiments or hypotheses that include human CD4^+^ T cells. Although proteins of interest may be present in heavily used T cell lines like Jurkat or non-T primary somatic cells, they may not be present in primary CD4^+^ T cells. 

## 8. Conclusions

CD4^+^ T cells are an integral part of the immune response, controlling the highly-specific effector immune functions of B cell antibody production, CD8^+^ T cytolytic ability, and macrophage activation. The evolutionary presence of CD4^+^ T cells alongside B and CD8^+^ T cells emphasize their critical role in enhancing immune responses against pathogens while protecting against allergies and autoimmunity. The clinical pathologies that arise during CD4^+^ T cell absence, such as during AIDS progression or in several primary immunodeficiencies, highlights their importance in human health. Understanding the polarized anatomy and temporal protein expression of CD4^+^ T cells allows us to acquire a deeper understanding and appreciation for these cells and their function. This understanding will be helpful to advance both pro- and anti-inflammatory immunotherapies that could decrease transplant rejections or treat a broad spectrum of disease ranging from allergies and autoimmunity to cancer and infectious diseases. 

## Figures and Tables

**Figure 1 biomedicines-11-02608-f001:**
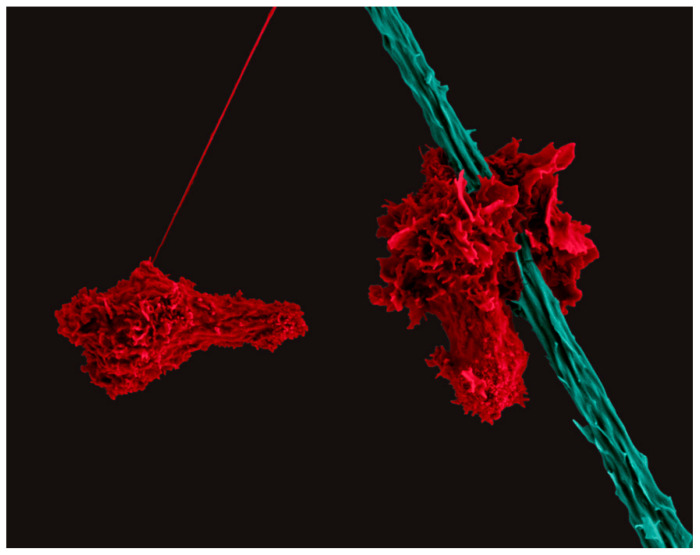
Scanning Electron Micrograph of two human CD4^+^ T cells (red) migrating with a polarized cell morphology. The T cell on the right scans a dendritic cell (teal). Created with BioRender.com.

**Figure 2 biomedicines-11-02608-f002:**
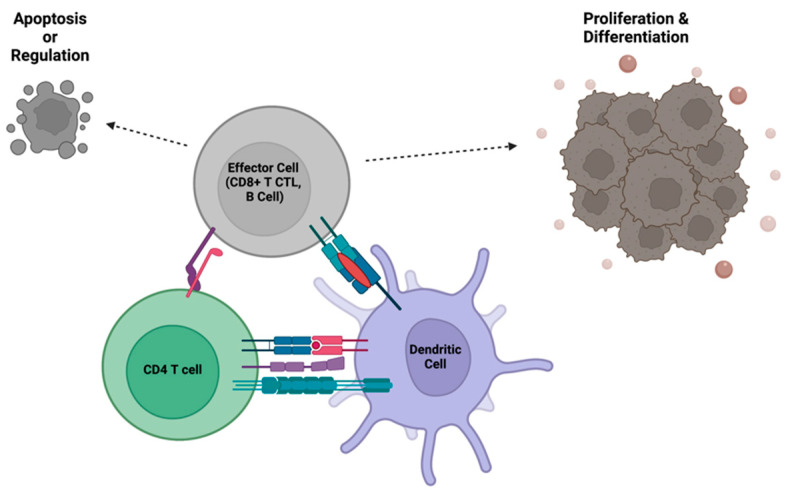
CD4^+^ T cells regulate the functions of effector immune cells, such as cytotoxic T cells and B cells, through direct interactions with both the effector cell and an antigen-presenting cell. The communication between CD4^+^ T cells and effector cells allows for clonal expansion and initiation of the adaptive immune response, or termination of effector functions after the pathogen has been cleared. Created with BioRender.com.

**Figure 3 biomedicines-11-02608-f003:**
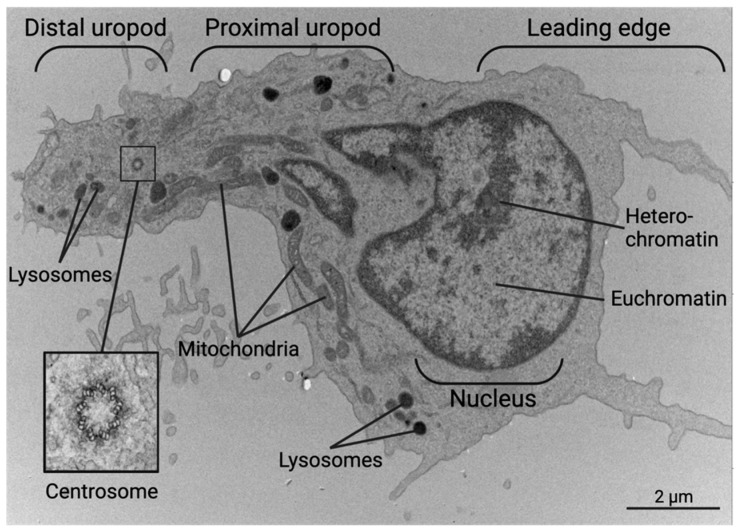
Transmission Electron Micrograph of a migrating human CD4^+^ T cell with a polarized morphology. The leading edge contains an actin-rich lamellipodium and filopodia while the uropod contains many organelles such as mitochondria, lysosomes, and rough ER on microtubule tracks. The centrosome separates the distal and proximal uropod in the trailing edge on the left. Created with BioRender.com.

**Figure 4 biomedicines-11-02608-f004:**
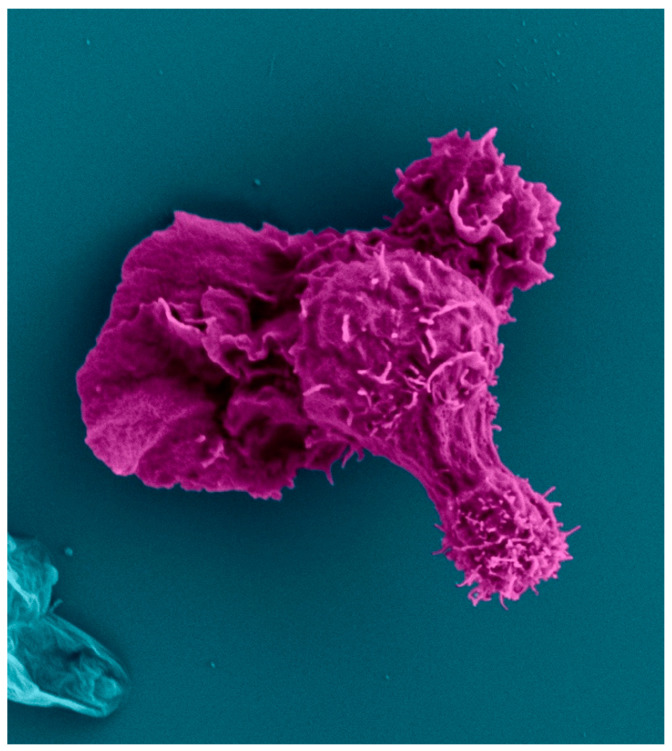
In this migrating CD4^+^ T cell, the leading edge, nucleus, and proximal and distal uropod all display different ruffling of the plasma membrane. This ruffling can be used to determine the directional movement of the cell. This particular cell is migrating to the left with a broad lamellipodial front. The retraction arm can be seen on the top right with the spherical nucleus connecting the two. The distal uropod or uropod “knob” can be seen as a distinct structure at the distal tip at the cell rear. Created with BioRender.com.

**Figure 5 biomedicines-11-02608-f005:**
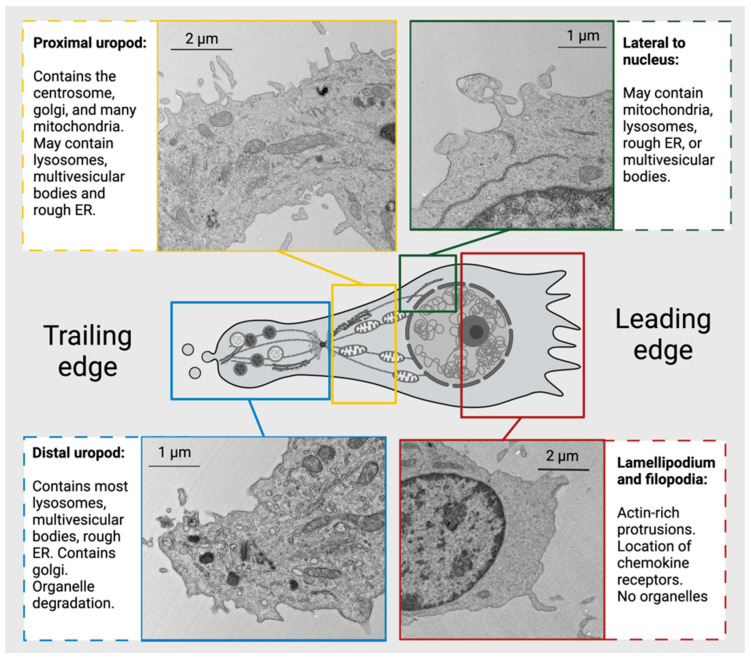
Organelles and cytoskeletal elements display subcellular localization in migrating CD4^+^ T cells. Most organelles are located posterior to the nucleus of a migrating cell, allowing the lamellipodia in front of the nucleus to guide cell movement. Centrosome localization divides the uropod into two segments: the distal and proximal uropod, which contain different organelle populations. Created with BioRender.com.

**Table 1 biomedicines-11-02608-t001:** Primary Immunodeficiencies Affecting T Cell Development and Function.

	Mutation	Cell Type(s) Affected	Mechanism of Immunodeficiency	Clinical Implications and Treatments	
** *Developmental deficiencies* **					
DiGeorge Syndrome	22q11.2 deletion	T cells	Small or no thymus, low T cell counts.	Hematopoietic stem cell transplant or thymus transplant (in infancy) may be necessary	[40,41]
Omenn Syndrome	Mutation in *RAG1* or *RAG2*	T and B cells	Diminished lymphocyte activation receptor variability, low B cell counts, defective negative selection in thymus (Normal-high T cell counts)	Elevated IgE levels, predisposition to autoimmunity	[42,43]
Adenosine Deaminase (ADA) deficiency	Limited to no ADA expression (ADA is normally expressed in the thymus at high levels)	All lymphocytes, but mainly T cells	Depletion of developing lymphocytes via toxic accumulation of 2′deoxyadenosine and 2′dioxyinoside	Can result in severe combined immunodeficiency (SCID), and increased susceptibility to viral infections.	[44]
Cartilage Hair Hypoplasia	*RMRP* gene mutation, important in RNA processing	All lymphocytes, but mainly T cells	Limited T cell maturation and differentiation, increased T cell apoptosis	Can result in SCID and a form of dwarfism, may have decreased antibody levels.	[45]
** *Receptor deficiencies leading to pathophysiological activation:* **					
CD25 deficiency	Mutation in *IL2RA* gene (α chain of IL-2 receptor)	T cells	Limited T cell development, proliferation, and activation, diminished IL-10 production (normal B cells)	Can result in SCID	[46,47]
X-linked lymphoproliferative syndrome-1 (XLP1)	Mutation in signaling lymphocyte-associated molecule (SLAM)-associated protein (SAP)	Lymphocytes	Limited T cell help and cytotoxicity, limited NK cell function	Increased incidence of lymphoma. HSCT is needed to cure	[48]
MHCII Deficiency (Bare lymphocyte syndrome)	MHCII genes intact, mutations in genes regulating MHC transcription	CD4^+^ T cells and APCs	Reduced CD4^+^ T cell counts due to incomplete maturation from perturbed positive and negative selection in thymus	Persistent viral infections. HSCT is needed to cure	[49]
Hyper IgM Syndrome	Mutation of CD40 on CD4^+^ T cells, or CD40L on B cells	CD4^+^ T cells and B cells	B cells cannot class switch out of IgM due to no CD40/CD40L interactions with CD4^+^ T cells	Increased bacterial infections, increased serum IgM levels. HSCT may be used to treat.	[50]
Chronic Mucocutaneous Candidiasis	Many causes, some include RORγT or IL-17 receptor deficiency	Th17 cells	Limited to no differentiation into Th17 cells and limited anti-fungal immunity	Chronic candida fungus infection, treatments may include antifungals. HSCT may be used to cure	[51]
** *Cytoskeletal defects that lead to pathophysiological migration and activation:* **					
Wiskott-Aldrich Syndrome	Dysfunctional Wiskott-Aldrich syndrome protein (WASp)	All lymphocytes, but mainly T cells	Inability of lymphocytes to create branched actin filaments, critical for immune cell migration and TCR activation.	Limited CD8^+^ T cell and B cell function, both from intrinsic defects and restricted CD4^+^ T cell help	[52,53]
Wiskott-Aldrich Syndrome-2	*WIPF1* gene mutation-WIP protein mutation (WASP-interacting protein)	Mainly T cells	Defective F-actin polymerization, leading to limited T cell migration and TCR activation. Low B and CD8^+^ T cell counts	Similar presentation to Wiskott-Aldrich Syndrome	[54]
DOCK2 deficiency	DOCK2	Hematopoietic cells, but mainly T cells	Limited Rac1 activation in T cells, reduced F actin polymerization	Possible decreased antibody production, decreased antiviral response. HSCT is needed to cure	[55]
DOCK8 Deficiency	Deficient DOCK8 protein (Normally, DOCK8 interacts with Cdc42, leading to branched actin creation)	All lymphocytes, but mainly T cells	Limited T cell migration, activation, and proliferation.	Severe allergic responses, elevated IgE levels, high risk for skin infections. HSCT is necessary to cure	[56]
NCKAP1 gene mutation	HEM1 protein (part of WAVE complex)	All immune cells, but mainly T cells and NK cells	Limited leading edge actin polymerization and migration, diminished immune synapse formation	Hyperinflammation, autoimmunity, recurring infections. May be treated with corticosteroids	[57,58]
ARPC1B Deficiency	ARPC1B (assists ARP2/3 complex)	Hematopoietic cells	No immune synapse formation in T cells, limited migration	Autoimmunity, combined immunodeficiency	[59,60]
CORO1A mutation	CORO1A C-terminal domain truncation	Hematopoietic cells, but mainly T cells	Inability for CORO1A to depolymerize actin cytoskeleton, leading to increased F-actin accumulation. Decreased T cell help	Limited CD4^+^ T cells, chronic viral infections, similar presentation to Wiskott-Aldrich syndrome	[61]
CDC42 Deficiency	CDC42	T cells and B cells	Impaired antibody production and T cell effectors function	Decline in T cell numbers and function, can treat some opportunistic infections with antibiotic prophylaxis	[62]
RAC2 Deficiency	RAC2	Hematopoietic cells	Decreased naïve CD4^+^ T cells, decreased neutrophil chemotaxis	Recurrent infections. HSCT can be used to cure	[63]

## Data Availability

The data presented in this study are available in the Appendix A.

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
