# Peer review of "A History and Atlas of the Human CD4+ T Helper Cell"

_biomedicines, 2023, doi:10.3390/biomedicines11102608_

Round 1
Reviewer 1 Report
The manuscript (review) by Crater et al., majorly focuses on CD4+ T cells and its evolution. Unfortunately, the review is unfocused, and there is no new interpretation has been discussed. Discussion of discovery section is very redundant and has no novel interpretation in context of CD4+ T cell functions. The discussion topics has been discussed in more than 50 reviews on CD4+T cells.
Specific comments
1.There are many recent interesting reviews are missing. Discussing the content of these reviews will provide the novelty to the manuscript about CD4+ T cells functional evolution. For example:
- The Era of Cytotoxic CD4 T Cells (Front. Immunol., 27 April 2022).
- CD4+ T cell memory (Nature Immunology
- The Many Faces of CD4+ T Cells: Immunological and Structural Characteristics (Int J Mol Sci. 2021 Jan; 22(1): 73.)
- Functionally specialized human CD4+ T-cell subsets express physicochemically distinct TCRs ( https://doi.org/10.7554/eLife.57063)
2. The abstract and the conclusion is very broad and not focused to the content of the manuscript.
3. The discussion of the entire review is very similar to the published reviews about CD4+ T cells. It needs to be refined according to functional characteristics of CD4+ T cells and its lineages.
Some sentences are too long and in passive voice. It would be nice if the sentences in active voice and easy for the non english speaking person to understand the content.
Author Response
"Please see the attachment."

Reviewer 2 Report
The authors provide a kind of story-telling historical review about human T helper cells.
I am missing a study concerning the 2-cell/3-cell interaction theory of CD8+T cell priming, maybe:
- Hoyer S, Prommersberger S, Pfeiffer IA, Schuler-Thurner B, Schuler G, Dörrie J, Schaft N. Concurrent interaction of DCs with CD4(+) and CD8(+) T cells improves secondary CTL expansion: It takes three to tango. Eur J Immunol. 2014 Dec;44(12):3543-59. doi: 10.1002/eji.201444477. Epub 2014 Oct 27. PMID: 25211552.
The distinction of the several functionally specialized CD4 T cell flavours (Th1/2/17/9 etc.) is not really discussed. Maybe some good review citations can be added.
Additionally, the role of CD4 help and Tregs in transplantation is not mentioned.
Maybe the authors should also shortly discuss the potential existence of CD4 "killer" T cells
However, the role of DCs for T cell priming is somewaht underrepresented (at least som citations or links to extensive reviews would be nice for the reader).
I would, for example propose some of those:
- Amon L, Hatscher L, Heger L, Dudziak D, Lehmann CHK. Harnessing the Complete Repertoire of Conventional Dendritic Cell Functions for Cancer Immunotherapy. Pharmaceutics. 2020 Jul 14;12(7):663. doi: 10.3390/pharmaceutics12070663. PMID: 32674488; PMCID: PMC7408110. - Amon L, Lehmann CHK, Baranska A, Schoen J, Heger L, Dudziak D. Transcriptional control of dendritic cell development and functions. Int Rev Cell Mol Biol. 2019;349:55-151. doi: 10.1016/bs.ircmb.2019.10.001. Epub 2019 Nov 15. PMID: 31759434. - León B, Lund FE. Compartmentalization of dendritic cell and T-cell interactions in the lymph node: Anatomy of T-cell fate decisions. Immunol Rev. 2019 May;289(1):84-100. doi: 10.1111/imr.12758. PMID: 30977197; PMCID: PMC6464380.
no comments
Author Response
"Please see the attachment."

Reviewer 3 Report
The authors have knowledgeably summarized the history of human CD4 T cell differentiation and biological function. The manuscript is focused and organized very well, which is interesting and helpful to understanding the role of human CD4 T cells in health and diseases. In addition, they also used their novel electro-microscopy data to verify how the cells migrate and function in vitro. All figures are presented well.
Minor concern and suggestion.
1. In line 239, after CD4+ T cells, the related figures/data should be indicated.
2. In line 251, English grammar needs to be checked.
3. Although the authors presented the primary data from transcriptomic and proteomic experiments, the analysis is relatively poor. The main signaling pathways in different stages of the activation need to be presented as figures.
Author Response
"Please see the attachment."

Round 2
Reviewer 1 Report
The authors adequately addressed the raised concerns in the first round of review process. The manuscript is significantly enhanced its quality. This satisfies the content of the manuscript to convey the scientific relevance to the field.